# Glucose-Lowering Effects and Safety of *Bifidobacterium longum* CKD1 in Diabetic Dogs and Cats

**DOI:** 10.3390/microorganisms13122881

**Published:** 2025-12-18

**Authors:** Yukyung Choi, Ji-Eun Kim, Kyung Hwan Kim, Sunghee Lee, Chang Hun Shin

**Affiliations:** Central Research Institute, Chong Kun Dang Bio, Ansan 15604, Republic of Korea; micro_yk@ckdbio.com (Y.C.); jieun.kim2025@ckdbio.com (J.-E.K.); kimkh@ckdbio.com (K.H.K.); slee@ckdbio.com (S.L.)

**Keywords:** diabetes mellitus, dogs, cats, *Bifidobacterium longum* CKD1

## Abstract

Diabetes mellitus is a common endocrine disorder in dogs and cats, and achieving stabilization with insulin alone can be challenging. This study evaluated the glucose-lowering efficacy and safety of *Bifidobacterium longum* CKD1 in diabetic companion animals. Nine dogs and 13 cats received *B. longum* CKD1 daily for 12 weeks. Fasting blood glucose (FBG) levels decreased by 26.2% in dogs and 18.0% in cats. Remarkable improvement in FBG was observed in dogs and cats with baseline levels exceeding 200 mg/dL in dogs and 250 mg/dL in cats. Notably, dogs exhibited a significant 61.8% reduction (*p* < 0.05), while cats showed a 38.4% decrease. Insulin-treated cats required significantly lower insulin doses by Week 12 (*p* < 0.05). Continuous glucose monitoring in cats demonstrated a 21.6% reduction in mean glucose levels and a significant 32.3% decrease in the time spent with glucose levels above 181 mg/dL (*p* < 0.05). Microbiota analysis revealed an increase in beneficial commensals and short-chain fatty acid producers, along with a reduction in pathobionts. No treatment-related adverse effects were observed. These findings indicate that *B. longum* CKD1 improves glycemic control and safely modulates the gut microbiota, supporting its potential use in managing diabetes mellitus in companion animals.

## 1. Introduction

Diabetes mellitus (DM) is one of the most common endocrine disorders in companion animals, particularly in middle-aged to senior dogs and cats, and is frequently associated with overweight and obesity [1,2]. In dogs, most cases resemble type 1 diabetes characterized by primary pancreatic β-cell failure and absolute insulin deficiency [3]. By contrast, diabetes in cats more closely resembles type 2 diabetes and typically develops through a multifactorial interaction between insulin resistance and β-cell dysfunction in the context of obesity, chronic pancreatitis, or other hormonal disorders [2,3]. Accordingly, insulin resistance is considered a central determinant of diabetes in cats [4]. In a subset of diabetic cats, hypersomatotropism secondary to pituitary adenomas causes excessive growth hormone secretion and increased circulating insulin-like growth factor-1 (IGF-1) concentrations, which in turn induce marked insulin resistance and represent an important endocrine cause of diabetes in cats [5].

The standard treatment approach, centered on exogenous insulin administration, dietary control, and weight management, is generally effective for glycemic regulation but is confounded by daily injections, the risk of hypoglycemia, and inconsistent owner adherence [1,6]. Furthermore, achieving sustained remission in clinical practice remains uncommon [7]. Recently, oral sodium-glucose cotransporter 2 (SGLT2) inhibitors have emerged as novel therapeutic options for feline diabetes. The U.S. Food and Drug Administration (FDA) approved bexagliflozin (Bexacat™, Elanco, Indianapolis, IN, USA) as the first non-insulin oral therapy for insulin-naive cats [8,9]. Subsequently, velagliflozin (Senvelgo^®^, Boehringer Ingelheim, Ingelheim am Rhein, Germany), another SGLT2 inhibitor formulated as an oral solution, received FDA approval, further expanding treatment options for feline diabetes. In Republic of Korea, enavogliflozin (DWP16001, Daewoong Pharmaceutical, Seoul, Republic of Korea), initially developed for the treatment of human type 2 diabetes, has demonstrated glycemic improvement in both pilot and randomized controlled clinical trials involving diabetic dogs [10]. Despite these advances, all currently available or investigated agents are small-molecule SGLT2 inhibitors. To date, no microbiome-based therapeutics have been developed or approved for managing diabetes in companion animals. Given the growing evidence that probiotics can modulate host glucose metabolism, microbiome-derived interventions may represent novel, safe, and complementary therapeutic strategies.

The gut microbiota has recently been recognized as a central modulator of metabolic and immune pathways in both canines and felines [11,12,13]. Microbial dysbiosis has been associated with impaired glycemic control and low-grade inflammation, with specific microbial taxa and metabolites, particularly short-chain fatty acids (SCFAs), implicated in these pathogenic processes [13,14]. In diabetic animals, both dogs and cats commonly exhibit reduced microbial diversity and a loss of butyrate-producing commensals [15,16], alongside further compositional shifts in cats with metabolic impairments [17]. Microbiome-targeted interventions such as dietary modification, prebiotics, synbiotics, and fecal microbiota transplantation, have been shown to remodel gut communities and metabolites in companion animals, with preliminary evidence suggesting potential benefits for metabolic outcomes [12,13,18,19]. However, systematic evaluation of live biotherapeutic products (LBPs), defined as viable microorganisms that confer health benefits, remains limited in veterinary medicine [20,21]. Within this class, *Bifidobacterium longum* has demonstrated robust effects in supporting intestinal barrier integrity, modulating immune responses, and improving glycemic control across various experimental systems [22,23]. Nonetheless, the translational potential of *B. longum* for clinical management of diabetes in companion animals remains underexplored.

This study investigated the glucose-lowering efficacy, safety, and microbiome-modulatory effects of *B. longum* CKD1 supplementation in diabetic dogs and cats. We hypothesized that daily administration of *B. longum* CKD1 would be both safe and beneficial, either as a monotherapy or as an adjunct to conventional treatment. To evaluate its efficacy, we assessed glycemic control through conventional and continuous glucose monitoring (CGM), insulin dosage adjustments, and fecal microbiota profiling, as well as functional metabolic pathway analyses. This integrated approach was designed to elucidate both the therapeutic efficacy and the underlying mechanisms of *B. longum* CKD1 supplementation in diabetic dogs and cats, potentially establishing its role as a novel, noninvasive management strategy.

## 2. Materials and Methods

### 2.1. Study Animals and Ethical Approval

This prospective, proof-of-concept, open-label study was approved by the Institutional Animal Care and Use Committee (IACUC) of Huvet Co., Ltd. (Jeongeup, Republic of Korea; approval numbers: HV2023-012 and HV2023-012-RENEW). The study was conducted between 2023 and 2025 across 19 veterinary clinics in Republic of Korea, including sites in Seoul, Bucheon, Yongin, Hwaseong, Anyang, Siheung, Seongnam, Iksan, Jeonju, and Gyeongsan. Written informed consent was obtained from all animal owners before enrollment.

A total of 13 client-owned dogs (all receiving insulin therapy) and 15 cats (12 insulin-treated and 3 non-insulin-treated) were initially enrolled. Diabetes mellitus was diagnosed in dogs with fasting blood glucose levels exceeding 200 mg/dL and in cats with fasting blood glucose levels exceeding 250 mg/dL, or based on the presence of glucosuria, elevated HbA1c, or fructosamine concentrations greater than 365 μmol/L, in accordance with established diagnostic criteria [1,24,25,26]. Animals were excluded if they had received lactic acid bacterial supplements within the previous month or had a history of other endocrine or metabolic disorders.

During the study, four dogs and two cats were excluded because they failed to maintain a fasting status or discontinued insulin therapy before blood sampling. Consequently, 9 insulin-treated dogs and 13 cats (10 insulin-treated and 3 non-insulin-treated) were included in the final analysis. Subgroup analyses were additionally performed for animals with baseline fasting blood glucose levels exceeding 200 mg/dL in dogs or 250 mg/dL in cats (Week 0) (Figure 1).

### 2.2. Administration of B. longum CKD1 and Measurement of Glycemic Parameters

*B. longum* CKD1 freeze-dried powder was administered daily for 12 weeks, mixed with the animals’ feed (1.0 × 10^10^ CFU/day for animals ≤ 7 kg; 2.0 × 10^10^ CFU/day for animals > 7 kg). To evaluate glucose-lowering effects, blood samples were collected from the cephalic or jugular vein after an overnight fast of at least 8 h, and urine samples were obtained via cystocentesis at baseline (Week 0) and at Weeks 6 and 12. Fasting blood glucose, glycated hemoglobin (HbA1c), fructosamine, and urine glucose levels were measured. HbA1c concentrations were determined using the CLOVER A1c Vet system (Green Cross Veterinary Products, Yongin, Republic of Korea), and fructosamine levels were analyzed using Green Vet Diagnostics (Republic of Korea). Insulin doses were recorded in insulin-treated animals to assess their effects on blood glucose regulation. CGM was performed at Weeks 0 and 12 to evaluate both the short- and long-term effects of *B. longum* CKD1 supplementation. The probiotic was administered for an additional week (Week 13) to standardize the treatment period. CGM was conducted using the FreeStyle Libre system (Abbott, Chicago, IL, USA), which was affixed to the skin, allowing glucose levels to be recorded for up to two weeks. Changes in blood glucose were analyzed, including the mean glucose concentration and the percentage of time spent within predefined glycemic ranges.

### 2.3. Gut Microbiota Analysis

Fecal samples were collected at Weeks 0 and 12 using sterile cotton swabs and stored at –80 °C until analysis. All procedures, including genomic DNA extraction, library preparation, sequencing, and bioinformatics analyses, were performed by Macrogen Inc. (Seoul, Republic of Korea). The V3–V4 region of the 16S rRNA gene, targeted by primers Bakt_341F-805R, was amplified and sequenced on an Illumina MiSeq platform (Illumina Inc., San Diego, CA, USA) in paired-end mode with a read length of 2 × 301 bp. Across all samples, sequencing yielded ~0.1–0.15 GB of sequence data per sample.

Data preprocessing and amplicon sequence variant (ASV) generation were performed following Macrogen’s standard pipeline. After sequencing, Cutadapt (v.3.2) was used to remove adapter and primer sequence from the raw reads, and forward and reverse reads were trimmed to 250 bp and 200 bp, respectively. Reads with expected errors of ≥ 2 were discarded. ASVs were then inferred using the DADA2 pipeline (v1.18.0), which performs error correction, denoising, merging of paired-end reads, and chimera removal. Taxonomic assignment was performed using the naïve Bayesian classifier implemented in DADA2 against the NCBI_16S rRNA reference database.

Microbial community composition was analyzed using QIIME (version 1.9.0 [27]) for downstream diversity and abundance assessments. Community diversity was evaluated using α-diversity indices (Shannon, Gini–Simpson, observed richness, and Faith’s phylogenetic diversity) and β-diversity metrics (Bray–Curtis, weighted, and unweighted UniFrac distances), which were visualized by principal coordinate analysis (PCoA) based on unweighted UniFrac distances. The mean relative abundance (%) of taxa at the genus or species level was compared between baseline (Week 0) and Week 12. Heatmaps were generated using GraphPad Prism (version 10.5.0; San Diego, CA, USA) to display taxa showing significant differences in relative abundance between time points. Taxa were included in the analysis if they met both of the following criteria: (i) *p* < 0.5 for differences in mean abundance, and (ii) a mean relative abundance ≥ 1% at either baseline (Week 0) or Week 12. Fold changes were calculated as the log_2_-transformed ratio of mean relative abundance between baseline and Week 12. Functional predictions of microbial communities were performed using the PICRUSt2 [28] pipeline based on KEGG Orthology (KO) annotations. Predicted KO and pathway-level abundances at baseline and Week 12 were compared using paired non-parametric test (Wilcoxon signed-rank tests), with *p* < 0.05 considered statistically significant.

### 2.4. Assessment of Hematological and Biochemical Parameters for Safety Monitoring

To monitor the safety profile of *B. longum* CKD1, complete blood count (CBC) and serum biochemical analyses were performed. Hematological parameters were measured using a Microsemi LC-662G hematology analyzer (Horiba, Kyoto, Japan), and serum biochemical parameters were analyzed using an Auto Chemistry Analyzer AS-280 (BIOELAB, Nanjing, China). CBC parameters included red blood cell (RBC), white blood cell (WBC), hemoglobin (Hb), platelet (PLT), and hematocrit (HCT) levels. Serum biochemistry parameters included total protein (TP), albumin (ALB), alkaline phosphatase (ALP), total bilirubin (TBIL), inorganic phosphorus (IP), total cholesterol (TC), γ-glutamyl transferase (GGT), alanine aminotransferase (ALT), calcium (Ca), creatinine (CRE), blood urea nitrogen (BUN), and globulin (GLOB). Body weight was also recorded to monitor general health status.

### 2.5. Statistical Analysis

The normality of data within each group was assessed prior to statistical testing. For normally distributed variables, paired *t*-tests were used to compare baseline (Week 0) values with those at Weeks 6 or 12. For non-normally distributed data, the Wilcoxon matched-pairs signed-rank test was applied. To further assess whether the observed glycemic improvements were influenced by baseline fasting glucose levels and to address the possibility of regression-to-the-mean bias, linear regression analyses were additionally performed between baseline FBG and the percentage change in FBG at Weeks 6 and 12. All statistical analyses were performed using GraphPad Prism software version 10.5.0 (GraphPad Software Inc., San Diego, CA, USA), and statistical significance was defined as *p* < 0.05.

## 3. Results and Discussion

### 3.1. Characteristics of Study Population

A total of 9 insulin-treated dogs and 13 cats (10 insulin-treated and 3 non-insulin-treated) were included in the final analysis. The canine cohort consisted of Poodle (*n* = 4), Maltese (*n* = 2), Miniature Pinscher (*n* = 1), Pomeranian (*n* = 1), and Shih Tzu–Poodle mix (*n* = 1). The median age of the dogs was 12 years (range: 8–14 years), and the median body weight was 5.0 kg (range: 4.0–6.7 kg). The feline cohort included Korean Shorthair (*n* = 6), Abyssinian (*n* = 2), Turkish Angora (*n* = 2), Russian Blue (*n* = 2), and Bengal (*n* = 1). Their median age was 11 years (range: 6–15 years), and the median body weight was 6.4 kg (range: 4.5–8.6 kg) (Table 1).

### 3.2. Glycemic Response to B. longum CKD1

Following administration of *B. longum* CKD1, fasting blood glucose levels at Week 12 decreased in both dogs (−26.2%) and cats (−18.0%), while urine glucose levels also declined in dogs (−46.3%) and cats (−18.4%) (Figure 2a,b). The greater reductions in fasting blood glucose and urine glucose observed in dogs compared with cats may be associated with differences in insulin dosing during the study. The relative insulin doses in insulin-treated dogs remained stable, whereas insulin-treated cats exhibited a significant decrease in insulin dose by Week 12 (*p* < 0.05; Figure 3). This variation in insulin administration may have contributed to the comparatively greater glycemic improvements observed in dogs. These findings suggest that if insulin doses have remained constant, the glycemic reduction in insulin-treated cats might have been even more pronounced.

HbA1c levels decreased in dogs at Week 6 (*p* = 0.0691) but did not show significant changes in either species by Week 12 (Figure 2c). No significant differences were observed in fructosamine levels throughout the study. HbA1c reflects average glycemia over approximately 2–3 months, whereas fructosamine represents average glycemia over 2–3 weeks [29,30]. In contrast, fasting blood glucose and urine glucose respond more rapidly to short-term fluctuations. Thus, the 12-week administration period of *B. longum* CKD1 may have been sufficient to improve immediate glycemic parameters but insufficient to induce measurable changes in intermediate- or long-term glycemic markers. Moreover, reductions in short-term indices may not have been consistent or sustained enough to influence these longer-term averages. These findings indicate that prolonged administration of *B. longum* CKD1 may be required to achieve continuous glycemic control.

In this study, some animals exhibited relatively low fasting blood glucose levels at baseline (Week 0) despite being diagnosed with diabetes based on other indices, such as urine glucose, HbA1c, or fructosamine. This discrepancy is likely attributable to ongoing insulin therapy prior to study enrollment. Because fasting blood glucose is considered the most reliable diagnostic marker for diabetes mellitus, with higher sensitivity and specificity than other indices [31], animals meeting the diagnostic threshold based on fasting blood glucose were subjected to additional analyses. In this subgroup of animals with higher baseline fasting blood glucose levels (dogs (*n* = 6) > 200 mg/dL and cats (*n* = 7) > 250 mg/dL), fasting blood glucose levels were significantly reduced at Week 12 in dogs (−61.8%, *p* < 0.05) and also decreased in cats (−38.4%) (Figure 4a). Urine glucose levels declined in both species (dogs, −35.8%; cats, −26.3%), and HbA1c levels were significantly reduced in dogs at Week 6 (−5.3%, *p* < 0.05) (Figure 4b,c). These results indicate that *B. longum* CKD1 exerts more pronounced glucose-lowering effects in animals with higher baseline glucose levels, suggesting potentially greater benefits in subjects with more severe hyperglycemia.

CGM was conducted at Weeks 0 and 12 to assess dynamic glycemic changes in diabetic cats. Mean blood glucose levels decreased by 21.6% (*p* = 0.0887), and the percentage of time spent above 181 mg/dL, a clinically relevant hyperglycemic threshold, was significantly reduced by 32.3% (*p* < 0.05) (Figure 5a,b). CGM enables the real-time assessment of blood glucose dynamics under the combined influence of diet, insulin therapy, and *B. longum* CKD1 supplementation, providing a more comprehensive evaluation of glycemic control than single-time-point measurements. These results demonstrate that *B. longum* CKD1 contributes to improved glycemic regulation under practical, real-world conditions.

### 3.3. Gut Microbiota Modulation by B. longum CKD1

Fecal microbiota profiling demonstrated that α-diversity metrics, including species richness and evenness, remained stable in both diabetic dogs and cats following *B. longum* CKD1 supplementation, indicating overall community stability [11,13]. In contrast, β-diversity analysis revealed clear compositional shifts, with treatment groups clustering distinctly from baseline samples (Figure 6a,b). These findings suggest that *B. longum* CKD1 induced targeted modifications in microbial composition rather than broad reductions in diversity.

Species-level differential abundance analysis identified significant increases in beneficial commensals and SCFA–producing taxa, including *Blautia hominis*, *B. longum* (the administered strain), *Faecalimonas umbilicata*, and *Megamonas funiformis*, within the canine gut microbiota (Figure 6c,d) [32,33,34,35,36]. Concurrently, several pathogenic taxa were significantly reduced, including *Escherichia fergusonii*, *Mediterraneibacter gnavus*, and *Enterococcus hirae* [37,38,39,40,41,42]. Notably, *B. longum* abundance increased from 0.00% to 6.82%, while *E. hirae* decreased from 4.00% to 0.80% and *M. gnavus* from 9.66% to 3.30% (Figure 6c,e).

In cats, microbial community patterns largely mirrored those observed in dogs, with increases in *B. longum* and other saccharolytic or SCFA-producing taxa, accompanied by decreases in established pathogenic species such as *Streptococcus canis* and *Helicobacter cinaedi* [33,43,44]. Specifically, *B. longum* abundance increased from 0.01% to 1.71%, while *S. canis* decreased from 3.24% to 1.51% and *H. cinaedi* from 2.84% to 1.35%. *Megamonas funiformis* also increased from 1.95% to 3.68% after 12 weeks (Figure 6d,f). Collectively, these compositional shifts reflected enrichment of taxa associated with intestinal health, metabolic regulation, and anti-inflammatory activity, alongside reductions in microorganisms linked to dysbiosis, impaired gut barrier function, and pro-inflammatory responses [32,38,45].

Functional pathway analysis in cats revealed significant post-intervention decreases in microbial pathways associated with steroid hormone biosynthesis and glycosaminoglycan degradation (Figure 6g). These alterations corresponded with the expansion of saccharolytic and SCFA-producing commensals and the decreased abundance of mucin- and glycosaminoglycan-degrading taxa [14,46]. Excessive microbial activation of steroid hormone biosynthesis has been implicated in altered gut permeability and heightened inflammatory signaling, compromising intestinal homeostasis [47,48,49]. Likewise, glycosaminoglycans are essential components of the intestinal mucus layer, and their microbial degradation can lead to mucosal injury, inflammation, and impaired gut integrity [46,47,50]. Given that aberrant steroid hormone metabolism and excessive glycosaminoglycan turnover have been linked to the pathophysiology of insulin resistance and diabetes [1], these functional modulations may underlie improved metabolic and intestinal stability.

Taken together, *B. longum* CKD1 supplementation remodeled the gut microbiota of diabetic dogs and cats toward a composition and functional state indicative of enhanced microbial homeostasis and reduced inflammatory potential. These outcomes were consistent across both taxonomic and predicted functional measures, reflecting host-specific yet concordant responses driven by probiotic administration and the baseline microbiome context [12,51].

Collectively, these microbiota-mediated alterations—including the enrichment of SCFA-producing taxa, the reduction of pro-inflammatory and mucin-degrading microorganisms, and the suppression of predicted microbial pathways related to intestinal injury and steroid hormone activation—suggest a shift toward a more favorable intestinal environment. These changes may contribute to improved gut barrier integrity and reduced intestinal inflammation. Such improvements in gut homeostasis have been linked to enhanced insulin sensitivity and glycemic stability in diabetes research [52,53]. In companion animals, decreased SCFA-producing bacteria have been reported in diabetic cats [15] and dogs with intestinal dysbiosis [16]. In contrast, we observed an increase in several SCFA-producing and saccharolytic taxa following *B. longum* CKD1 supplementation, a pattern that is consistent with the hypothesis that shifts in microbial composition may influence metabolic regulation.

Overall, the observed remodeling of the gut microbiota by *B. longum* CKD1 in this study suggests a plausible mechanistic pathway whereby improved intestinal health may indirectly modulate glucose metabolism in diabetic dogs and cats. However, while these findings provide preliminary mechanistic insights, further studies incorporating direct assessments of gut permeability, inflammatory biomarkers, and host metabolic signaling will be necessary to elucidate the molecular mechanisms by which *B. longum* CKD1 exerts its glycemic benefits.

### 3.4. Safety of B. longum CKD1

Evaluation of CBC and serum biochemistry parameters following *B. longum* CKD1 administration indicated no evidence of treatment-related toxicity. Although TP, GGT, and GLOB levels in dogs, as well as CRE levels in cats, were slightly higher at Week 12 compared with baseline (Week 0), these changes were not statistically significant (Table 2). Furthermore, no adverse clinical signs or significant changes in body weight were observed in any dog or cat throughout the study period (Figure 7).

## 4. Conclusions

In conclusion, this non-controlled, proof-of-concept clinical study demonstrated that *B. longum* CKD1 was well-tolerated, and showed promising signals of glycemic improvement in diabetic dogs or cats. These preliminary findings provide concept-level evidence that *B. longum* CKD1 may represent a safe and effective therapeutic option, either as a noninvasive monotherapy or as an adjunct to conventional insulin therapy, for managing diabetes mellitus in companion animals.

However, this study had certain limitations, including its relatively short duration and the absence of a placebo-controlled group. Due to the relatively short duration of the study, the long-term persistence of the observed glycemic improvements and the comprehensive safety profile, including potential treatment-related adverse effects, could not be fully evaluated. Therefore, further research involving long-term administration, larger sample sizes, and controlled comparative designs is warranted to confirm the sustained efficacy and long-term safety of *B. longum* CKD1.

Overall, the results support the potential of *B. longum* CKD1 as a novel, well-tolerated intervention for managing diabetes mellitus in dogs and cats, providing a foundation for further research aimed at its clinical application in veterinary medicine.

## Figures and Tables

**Figure 1 microorganisms-13-02881-f001:**
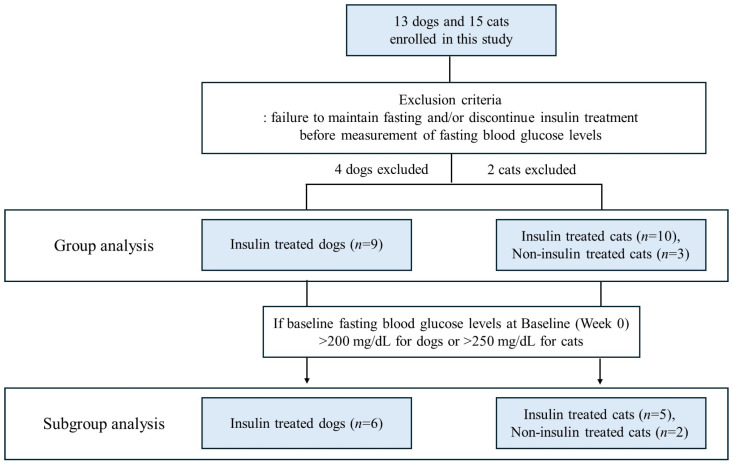
Flow diagram illustrating the group composition of diabetic dogs and cats enrolled in the study. Initially, 13 dogs and 15 cats were enrolled. After excluding subjects that failed to maintain fasting status or discontinued insulin therapy before fasting blood glucose measurement, data from 9 dogs and 13 cats were included in the analysis. Group analyses were performed for 9 insulin-treated dogs and 13 cats (10 insulin-treated and 3 non-insulin-treated). Subgroup analyses were conducted for dogs with baseline fasting blood glucose levels exceeding 200 mg/dL (*n* = 6) and for cats exceeding 250 mg/dL (*n* = 7) at Week 0.

**Figure 2 microorganisms-13-02881-f002:**
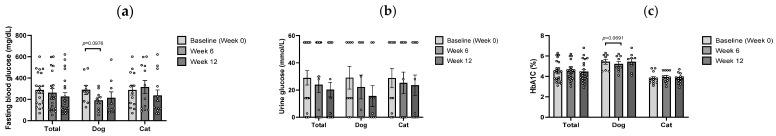
Fasting blood glucose (**a**), urine glucose (**b**), and glycated hemoglobin (HbA1c) (**c**) levels in diabetic dogs (*n* = 9) and cats (*n* = 13) at baseline (Week 0) and after 6 and 12 weeks of *Bifidobacterium longum* CKD1 administration.

**Figure 3 microorganisms-13-02881-f003:**
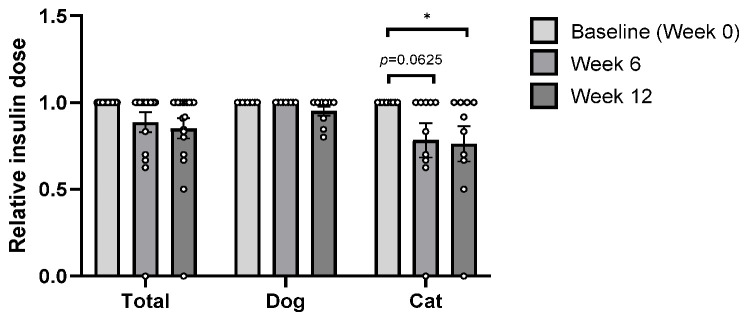
Relative insulin dose in diabetic dogs (*n* = 9) and cats (*n* = 10) at baseline (Week 0) and after 6 and 12 weeks of *Bifidobacterium longum* CKD1 administration. * *p* < 0.05 compared with baseline (Week 0), based on paired *t*-tests or Wilcoxon matched-pairs signed-rank tests.

**Figure 4 microorganisms-13-02881-f004:**
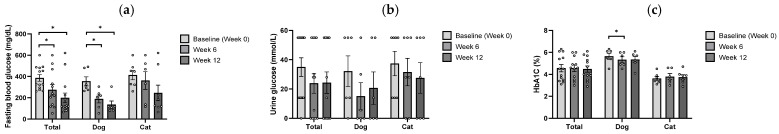
Fasting blood glucose (**a**), urine glucose (**b**), and glycated hemoglobin (HbA1c) (**c**) levels in diabetic dogs with baseline fasting blood glucose levels > 200 mg/dL (*n* = 6) and cats with levels > 250 mg/dL (*n* = 7) at baseline (Week 0) and after 6 and 12 weeks of *Bifidobacterium longum* CKD1 administration. * *p* < 0.05 compared with baseline (Week 0), based on paired *t*-tests or Wilcoxon matched-pairs signed-rank tests.

**Figure 5 microorganisms-13-02881-f005:**
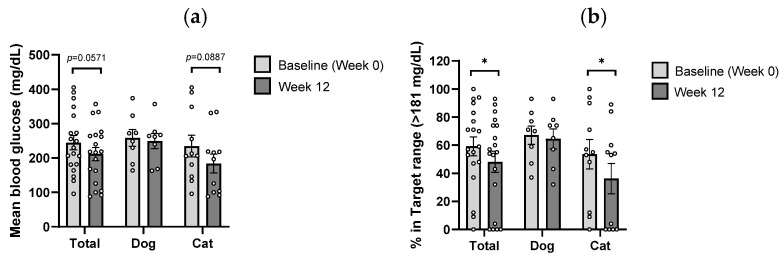
Continuous glucose monitoring (CGM) results in diabetic dogs (*n* = 8) and cats (*n* = 11) at baseline (Week 0) and after 12 weeks of *Bifidobacterium longum* CKD1 administration: (**a**) mean blood glucose levels and (**b**) percentage of time spent above the hyperglycemic threshold (> 181 mg/dL). * *p* < 0.05 compared with baseline (Week 0), based on paired *t*-tests or Wilcoxon matched-pairs signed-rank tests.

**Figure 6 microorganisms-13-02881-f006:**
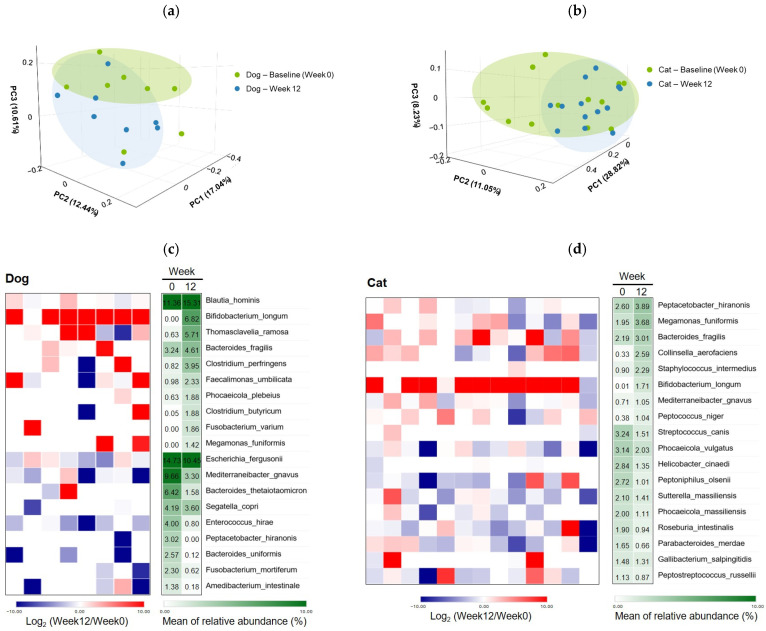
Modulation of the fecal microbiota by *Bifidobacterium longum* CKD1 in diabetic dogs (*n* = 9) and cats (*n* = 13). (**a**,**b**) Principal coordinate analysis (PCoA) plots of β-diversity based on unweighted UniFrac distances in dogs (**a**) and cats (**b**). (**c**,**d**) Heatmaps of bacterial taxa showing significant changes in fecal abundance in dogs (**c**) and cats (**d**) from baseline (Week 0) to post-treatment (Week 12). Only taxa meeting both of the following criteria are shown: (i) *p* < 0.5 for differences in mean abundance between time points; and (ii) taxa with mean relative abundance ≥ 1% at either baseline or Week 12, depending on the direction of change. The blue–red heatmaps depict log_2_-transformed fold changes in relative abundance (red, increase; blue, decrease). The green heatmaps indicate mean relative abundance (%) at baseline and Week 12, with darker green denoting higher abundance. (**e**,**f**) Relative abundance of selected taxa at Week 0 and Week 12 in dogs (**e**) and cats (**f**). (**g**) Predicted microbial functional pathways in cats inferred using PICRUSt2 based on KEGG Orthology (KO) annotations. * *p* < 0.05 and ** *p* < 0.01 compared with baseline (Week 0), based on paired *t*-tests or Wilcoxon matched-pairs signed-rank tests.

**Figure 7 microorganisms-13-02881-f007:**
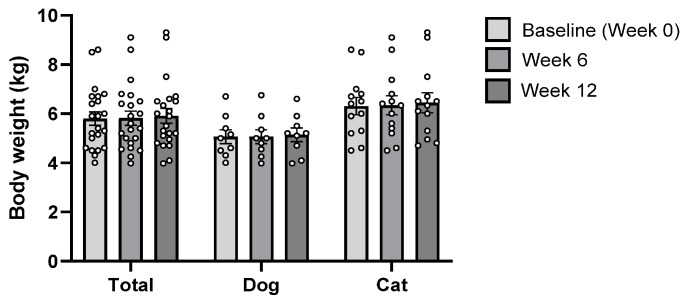
Changes in body weight in diabetic dogs (*n* = 9) and cats (*n* = 13) at baseline (Week 0) and after 6 and 12 weeks of *Bifidobacterium longum* CKD1 administration.

**Table 1 microorganisms-13-02881-t001:** Characteristics of the study population of diabetic dogs and cats.

Variables	Dogs (*n* = 9)	Cats (*n* = 13)
Breed	Poodle (*n* = 4), Maltese (*n* = 2), Miniature Pinscher (*n* = 1), Pomeranian (*n* = 1), Shih Tzu-Poodle Mix (*n* = 1)	Korean Shorthair (*n* = 6), Abyssinian (*n* = 2), Turkish Angora (*n* = 2), Russian Blue (*n* = 2), Bengal (*n* = 1)
Age, years Median (Min–Max)	12 (8–14)	11 (6–15)
Sex	Castrated male (4), Castrated female (5)	Castrated male (11), Castrated female (2)
Body weight, kg Median (Min–Max)	5.0 (4.0–6.7)	6.4 (4.5–8.6)

**Table 2 microorganisms-13-02881-t002:** Complete blood count and serum biochemistry in diabetic dogs (*n* = 9) and cats (*n* = 13) enrolled in this study at baseline (Week 0), Week 6, and Week 12 of *Bifidobacterium longum* CKD1 administration.

Category	Parameters	Unit	Dogs				Cats			
			Reference Range	Baseline (Week 0)	Week 6	Week 12	Reference Range	Baseline (Week 0)	Week 6	Week 12
Complete Blood Count (CBC)	WBC	10^9^/L	5.5–19.5	11.4 ± 3.1	9.0 ± 2.4	11.4 ± 5.2	5.5–19.5	17.3 ± 6.6	16.3 ± 7.3	15.9 ± 8.2
	RBC	10^12^/L	4.6–10	6.8 ± 1.1	6.7 ± 1.2	6.7 ± 1.2	4.6–10.0	8.4 ± 1.9	8.2 ± 1.4	8.6 ± 1.6
	HGB	g/dL	9.3–15.3	15.3 ± 2.3	15.6 ± 2.5	15.7 ± 2.6	9.3–15.3	13.3 ± 2.7	13.5 ± 2.1	13.8 ± 2.3
	HCT	%	28–49	46.2 ± 6.9	46.0 ± 8.6	46.8 ± 9.0	28–49.0	42.6 ± 8.3	41.5 ± 6.2	43.6 ± 6.2
	PLT	10^9^/L	100–514	371.6 ± 172.0	408.8 ± 157.6	378.7 ± 143.6	100–514.0	200.1 ± 172.4	190.7 ± 95.4	194.6 ± 105.4
Serum Biochemistry	TP	g/dL	5.5–7.2	7.2 ± 0.8	7.0 ± 0.7	12.9 ± 18.2	6.6–8.4	8.4 ± 1.0	8.4 ± 0.7	8.2 ± 0.7
	ALB	g/dL	3.2–4.1	3.6 ± 0.4	3.3 ± 0.3	3.4 ± 0.2	3.2–4.3	3.7 ± 0.6	3.9 ± 0.3	3.8 ± 0.4
	ALP	U/L	7–115	590.3 ± 606.1	522.4 ± 420.9	589.9 ± 511.5	11–49.0	47.6 ± 19.6	43.0 ± 26.4	50.4 ± 26.6
	TBIL	mg/dL	0–0.2	0.1 ± 0.1	0.1 ± 0.1	0.1 ± 0.1	0.1–0.5	0.1 ± 0.1	0.2 ± 0.1	0.1 ± 0.0
	IP	mg/dL	2.7–5.4	4.4 ± 0.7	4.1 ± 1.1	4.4 ± 0.9	2.6–5.5	4.8 ± 0.9	4.5 ± 1.2	4.2 ± 1.5
	TC	mg/dL	136–392	396.2 ± 165.5	409.9 ± 180.0	462.7 ± 257.5	101–323.0	231.5 ± 65.6	244.4 ± 89.9	243.5 ± 81.1
	GGT	U/L	0–8	8.6 ± 6.2	7.5 ± 8.7	15.9 ± 12.0	0–2.0	3.0 ± 1.4	7.2 ± 4.9	3.7 ± 2.5
	ALT	U/L	17–95	168.8 ± 175.4	102.0 ± 78.7	154.6 ± 137.4	22–84.0	90.1 ± 56.6	77.1 ± 45.0	71.2 ± 33.0
	Ca	mg/dL	9.4–11.1	10.7 ± 0.6	10.6 ± 1.4	9.9 ± 0.5	9–11.3	10.7 ± 0.8	9.7 ± 1.8	10.0 ± 0.6
	CRE	mg/dL	0.6–1.4	0.7 ± 0.3	0.7 ± 0.5	0.5 ± 0.3	0.8–2.1	1.6 ± 0.6	1.7 ± 0.6	3.6 ± 7.5
	BUN	mg/dL	9–26	26.2 ± 16.7	26.8 ± 16.7	28.5 ± 16.8	17–35.0	34.0 ± 10.1	37.0 ± 12.0	28.9 ± 11.6
	GLOB	g/dL	1.9–3.7	3.6 ± 0.4	3.7 ± 0.5	9.5 ± 18.3	2.9–4.7	4.7 ± 0.6	4.6 ± 0.6	4.4 ± 0.5

Abbreviations: WBC, white blood cell count; RBC, red blood cell count; HGB, hemoglobin; HCT, hematocrit; PLT, platelet; TP, total protein; ALB, albumin; ALP, alkaline phosphatase; TBIL, total bilirubin; IP, inorganic phosphorus; TC, total cholesterol; GGT, γ-glutamyl transferase; ALT, alanine aminotransferase; Ca, calcium; CRE, creatinine; BUN, blood urea nitrogen; GLOB, globulin.

## Data Availability

All 16S rRNA sequencing data supporting the current work are freely available in the European Nucleotide Archive (ENA) under the BioProject accession number PRJEB100620.

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
