# Peer review of "Glucose-Lowering Effects and Safety of *Bifidobacterium longum* CKD1 in Diabetic Dogs and Cats"

_microorganisms, 2025, doi:10.3390/microorganisms13122881_

Round 1
Reviewer 1 Report
Comments and Suggestions for Authors
-
Justify why a placebo or control group was not used. The absence of a comparator limits the ability to attribute the observed glucose-lowering effects solely to B. longum CKD1. Please clarify the study design and justify the lack of a control arm if applicable.
-
The sample size (9 dogs and 13 cats) appears small to draw strong conclusions regarding efficacy and safety. A power calculation or justification for the chosen sample size should be provided to strengthen the validity of the results.
-
The baseline fasting glucose levels differ significantly between animals. It would be helpful to include subgroup analyses or adjust for baseline glucose variability to ensure the improvements are not due to regression to the mean.
-
The microbiota-related findings are promising, but the manuscript does not provide sufficient detail on sequencing methods, diversity metrics, or statistical analyses. Please expand the methodology and present compositional changes with appropriate visualizations.
-
The study evaluates outcomes for 12 weeks, which may be insufficient to assess long-term safety and sustainability of glycemic improvements. Consider including a follow-up phase or discussing the implications of the relatively short duration.
-
While the study demonstrates significant glucose-lowering effects, the proposed mechanisms by which B. longum CKD1 influences glycemic control in dogs and cats are not discussed. Including mechanistic explanations or referencing prior research would improve the scientific depth and translational relevance.
Author Response
RESPONSES TO REVIEWER’S COMMENTS
The authors thank the editors and reviewers for constructive comments. Our responses to each of the comments are provided below.
Reviewer(s)' Comments to Author:
Reviewer 1
- Justify why a placebo or control group was not used. The absence of a comparator limits the ability to attribute the observed glucose-lowering effects solely to longum CKD1. Please clarify the study design and justify the lack of a control arm if applicable.
Response : We appreciate the reviewer’s valuable comment and agree that limitation of the absence of a placebo (control) group. Initially, our study was designed to include both a placebo-controlled group and a B. longum CKD1-treated group in newly diagnosed animals. However, it was challenging to recruit an adequate number of client-owned animals. Due to practical constraints that precluded extending the recruitment period indefinitely, we determined that the study would proceed without a placebo group and expanded the inclusion criteria to enroll not only newly diagnosed diabetic animals but also previously diagnosed, insulin-treated diabetic animals. We re-designed this study as a proof-of-concept evaluation and the animal study protocol was approved by the Institutional Animal Care and Use Committee (IACUC) of Huvet Co., Ltd. (approval numbers: HV 2023-012 (2023.10.30) and HV 2023-012-RENEW(2024.10.21).
In reference to Benedict et al. (2022), only five client-owned cats were evaluated to investigate the effects of bexagliflozin—an initially approved therapy for type 2 diabetic cats—without inclusion of a placebo group, which was acknowledged as a study limitation. An et al. (2024) investigated the glycaemic control effects of DWP16001 in diabetic dogs (n = 9–10 per group) in a pilot study, also without incorporating a placebo group. Accordingly, we explicitly acknowledged the absence of a placebo control group as a study limitation and tempered the strength of our conclusions in the “Conclusions” section (lines 350–352 and 357–360). The inclusion of a placebo control group could be considered to ensure a more rigorous assessment of B. longum CKD1 if we plan further robust clinical studies.
*Reference
- Benedict, S. L., Mahony, O. M., McKee, T. S., & Bergman, P. J. (2022). Evaluation of bexagliflozin in cats with poorly regulated diabetes mellitus. Canadian Journal of Veterinary Research, 86(1), 52-58.
- An, J. H., Choi, H. S., Choi, J. S., Lim, H. W., Huh, W., Oh, Y. I., ... & Youn, H. Y. (2024). Effect of the sodium‐glucose cotransporter‐2 inhibitor, DWP16001, as an add‐on therapy to insulin for diabetic dogs: A pilot study. Veterinary Medicine and Science, 10(3), e1454.
- The sample size (9 dogs and 13 cats) appears small to draw strong conclusions regarding efficacy and safety. A power calculation or justification for the chosen sample size should be provided to strengthen the validity of the results.
Response : We acknowledge that our sample size was limited. As noted in the response to the reviewer’s comment No. 1, our investigation was designed as a proof-of-concept trial in client-owned animals, focusing primarily on evaluating preliminary efficacy and feasibility rather than confirmatory outcomes. This limitation has been addressed and tempered the strength of our conclusions in the “Conclusions” section (lines 350–352 and 357–360).
- The baseline fasting glucose levels differ significantly between animals. It would be helpful to include subgroup analyses or adjust for baseline glucose variability to ensure the improvements are not due to regression to the mean.
Response : We appreciate the reviewer’s important comment. To address the reviewer’s concern regarding baseline variability and potential regression to the mean, we performed linear regression analyses between baseline fasting glucose levels and the percentage change in FBG at Weeks 6 and 12. Except for the total dataset and the dog-only dataset at Week 12 in Fig. 2a, all other regression slopes in Fig. 2a, 4a, 5a, and 5b were not significantly different from zero (p = 0.054 – 0.908), indicating that the treatment response did not depend on baseline glycemia. Importantly, in the datasets that demonstrated statistically significant glycemic improvements by paired t-tests, the regression slopes also remained non-significant (p = 0.11–0.73), providing direct evidence that the observed improvement were not attributable to regression-to-the-mean effects. This statistical method was described in the “2.4. Statistical Analysis” section of the revised manuscript (lines 179–182).
- The microbiota-related findings are promising, but the manuscript does not provide sufficient detail on sequencing methods, diversity metrics, or statistical analyses. Please expand the methodology and present compositional changes with appropriate visualizations.
Response : We thank the reviewer for this helpful comment. In the revised manuscript, we have substantially expanded the “2.3. Gut Microbiota Analysis” subsection of the methods to provide clearer information on the sequencing workflow, diversity metrics, and statistical approaches. Specifically, in lines 146–157, 162–163, and 172–174, we provide additional detail on the 16S rRNA sequencing workflow as well as the preprocessing/ASV pipeline. We more explicitly descrived that the criteria and statistical approach for compositional and functional analyses. We believe these additions address the reviewer’s concerns.
- The study evaluates outcomes for 12 weeks, which may be insufficient to assess long-term safety and sustainability of glycemic improvements. Consider including a follow-up phase or discussing the implications of the relatively short duration.
Response : We agree that a 12-week study period may be insufficient to fully assess long-term safety and the durability of glycemic improvements given the proof-of-concept of this study design. In response, we have added a statement in the Conclusion section explicitly acknowledging this limitation and addressing the need for future long-term studies to substantiate the sustained therapeutic potential and safety of B. longum CKD1 (lines 350–352 and 357–360).
- While the study demonstrates significant glucose-lowering effects, the proposed mechanisms by which longum CKD1 influences glycemic control in dogs and cats are not discussed. Including mechanistic explanations or referencing prior research would improve the scientific depth and translational relevance.
Response : Thank you for your constructive comments. We described the mechanistic explanation and the limitation of the results in the “Results and Discussion - 3.3. Gut Microbiota Modulation by B. longum CKD1” section (lines 314–331).
Reviewer 2 Report
Comments and Suggestions for Authors
The topic is very relevant for the field of diabetology. The authors have studied the glucose-lowering effects and safety of Bifidobacterium longum ckd1 in diabetic dogs and cats.
The methodology is very modern and complex, authors using modern methods of blood biochemistry, and bioinformatics analyses of gut microbiota. To monitor the safety profile of B. longum CKD1, complete blood count (CBC) and serum biochemical analyses were performed.
The results have revealed that the greater reductions in fasting blood glucose and urine glucose observed in dogs compared with cats, which may be associated with differences in insulin dosing during the study. The 12-week administration period of B. longum CKD1 may be sufficient to improve immediate glycemic parameters but insufficient to induce measurable changes in intermediate- or long-term glycemic markers. These results could be transferable to human health.
The conclusions are consistent with the evidence and arguments presented.
The references are very relevant, including also some relevant author’s previous experience in the field.
I suggest some corrections. For Introduction and Discussions authors may see also:
In Introduction authors should discuss also about insulinresistance. May see also Madalina Rosca, Yaiza Forcada, Gheorghe Solcan, David B Church and Stijn JM Niessen, 2014, Screening diabetic cats for hypersomatotropism: performance of an enzyme-linked immunosorbent assay for insulin-like growth factor 1, Journal of Feline Medicine and Surgery 2014 16: 82-88, DOI: 10.1177/1098612X13496246
Nr of cases is quite small, and data are collected from many clinics, so how can authors could be sure that experimental conditions are similar?
Author Response
RESPONSES TO REVIEWER’S COMMENTS
The authors thank the editors and reviewers for constructive comments. Our responses to each of the comments are provided below.
Reviewer(s)' Comments to Author:
Reviewer 2
- In Introduction authors should discuss also about insulinresistance. May see also Madalina Rosca, Yaiza Forcada, Gheorghe Solcan, David B Church and Stijn JM Niessen, 2014, Screening diabetic cats for hypersomatotropism: performance of an enzyme-linked immunosorbent assay for insulin-like growth factor 1, Journal of Feline Medicine and Surgery 2014 16: 82-88, DOI: 10.1177/1098612X13496246
Response : We agree that insulin resistance is an important component of the pathophysiology of diabetes in cats and should be explicitly addressed in the Introduction. In the revised manuscript, we have expanded the first phragraph of the introduction to (i) contrast the predominantly type 1-like form of diabetes in dogs with the type 2-like , insulin-resistant form in cats, and (ii) describe hypersomatotropism as an important endocrine cause of insulin-resistant diabetes in cats as the reviewer’s suggestion. We have cited the suggested article by Rosca et al. (Journal of Feline Medicine and Surgery 2014;16:82–88), which demonstrates the utility of serum IGF-1 measurement for screening diabetic cats for hypersomatotropism (lines 28–38). We believe that these additions provide a clearer clinical and pathophysiological context for our study.
- Nr of cases is quite small, and data are collected from many clinics, so how can authors could be sure that experimental conditions are similar?
Response : We appreciate the reviewer’s valuable comment and acknowledge that the sample size was limited. As this study was conducted as a proof-of-concept trial in client-owned animals, our primary objective was to explore preliminary efficacy and feasibility rather than to draw definitive conclusions. This approach is consistent with previously published studies in companion animal diabetes. For example, Benedict et al. (2022) evaluated only five client-owned cats to assess the effects of bexagliflozin—an FDA-approved therapy for type 2 diabetic cats—and recognized the limited sample size as a study constraint. Similarly, An et al. (2024) investigated the glycemic control effects of DWP16001 in diabetic dogs with approximately nine to ten animals per group in a pilot study. In alignment with such proof-of-concept investigations, we acknowledged the limitation of the sample size and have cautiously interpreted the findings in the “Conclusions” section (lines 350–352 and 357–360). Future controlled clinical studies may be considered to more robustly validate the therapeutic potential of B. longum CKD1.
*Reference
- Benedict, S. L., Mahony, O. M., McKee, T. S., & Bergman, P. J. (2022). Evaluation of bexagliflozin in cats with poorly regulated diabetes mellitus. Canadian Journal of Veterinary Research, 86(1), 52-58.
- An, J. H., Choi, H. S., Choi, J. S., Lim, H. W., Huh, W., Oh, Y. I., ... & Youn, H. Y. (2024). Effect of the sodium‐glucose cotransporter‐2 inhibitor, DWP16001, as an add‐on therapy to insulin for diabetic dogs: A pilot study. Veterinary Medicine and Science, 10(3), e1454.
In addition, we appreciate the reviewer’s important concern regarding potential variability across multiple veterinary clinics. We recognize that variations in the home environment and daily care of client-owned animals cannot be fully controlled, which remains an inherent limitation of real-world clinical studies. To minimize clinic-to-clinic differences, all procedures were standardized under the supervision of the same CRO (contract research organization), and the same personnel provided unified training to both participating veterinarians and pet owners. Consistent instructions were given regarding dosing administration, continuous glucose monitoring device attachment, and standardized visit requirements at Weeks 6 and 12, including fasting status and withholding insulin administration prior to sampling. Furthermore, blood samples were processed and analyzed at the same accredited testing facility using identical analytical instrumentation.
Round 2
Reviewer 2 Report
Comments and Suggestions for Authors
The authors have made all the corrections recommended. I recommend the acceptance of the article in this revised form
Author Response
We sincerely thank the reviewers for carefully evaluating the revised manuscript and for recommending its acceptance.